# *FOXE1*-Dependent Regulation of Macrophage Chemotaxis by Thyroid Cells In Vitro and In Vivo

**DOI:** 10.3390/ijms22147666

**Published:** 2021-07-17

**Authors:** Sara C. Credendino, Marta De Menna, Irene Cantone, Carmen Moccia, Matteo Esposito, Luigi Di Guida, Mario De Felice, Gabriella De Vita

**Affiliations:** 1Department of Molecular Medicine and Medical Biotechnology, University of Naples Federico II, 80131 Naples, Italy; saracarmela.credendino@unina.it (S.C.C.); marta.demenna@me.com (M.D.M.); irene.cantone@unina.it (I.C.); Carmen.moccia26@gmail.com (C.M.); matteo.esposito@unina.it (M.E.); luigi.diguida@unina.it (L.D.G.); mario.defelice@unina.it (M.D.F.); 2Institute of Experimental Endocrinology and Oncology “G. Salvatore”, National Research Council (CNR), 80131 Naples, Italy; 3DBMR-Department for BioMedical Research, University of Bern, 3012 Bern, Switzerland

**Keywords:** FOXE1, chemokines, TAMs, tumour microenvironment

## Abstract

Forkhead box E1 (*FOXE1*) is a lineage-restricted transcription factor involved in thyroid cancer susceptibility. Cancer-associated polymorphisms map in regulatory regions, thus affecting the extent of gene expression. We have recently shown that genetic reduction of FOXE1 dosage modifies multiple thyroid cancer phenotypes. To identify relevant effectors playing roles in thyroid cancer development, here we analyse *FOXE1*-induced transcriptional alterations in thyroid cells that do not express endogenous *FOXE1*. Expression of FOXE1 elicits cell migration, while transcriptome analysis reveals that several immune cells-related categories are highly enriched in differentially expressed genes, including several upregulated chemokines involved in macrophage recruitment. Accordingly, *FOXE1*-expressing cells induce chemotaxis of co-cultured monocytes. We then asked if *FOXE1* was able to regulate macrophage infiltration in thyroid cancers in vivo by using a mouse model of cancer, either wild type or with only one functional *FOXE1* allele. Expression of the same set of chemokines directly correlates with *FOXE1* dosage, and pro-tumourigenic M2 macrophage infiltration is decreased in tumours with reduced *FOXE1*. These data establish a novel link between *FOXE1* and macrophages recruitment in the thyroid cancer microenvironment, highlighting an unsuspected function of this gene in the crosstalk between neoplastic and immune cells that shape tumour development and progression.

## 1. Introduction

The incidence of thyroid cancer is constantly increasing, with the most frequent histotype, papillary thyroid carcinoma (PTC), representing 2% of total diagnosed neoplasms [1] and being one of the most common cancers in women under the age of 50 [2,3]. Although the differentiated forms, including PTC, have a good 5-year prognosis in 85% of cases [3], 15% remain with a poor prognosis and can either metastasise or progress to an aggressive anaplastic thyroid carcinoma (ATC). Despite mutations in several genes, as well as the expression levels of different markers having been associated with a poor prognosis, there are not yet defined molecular markers that could predict cancer progression [4]. Cancer cells are embedded in a complex microenvironment in which pivotal roles are played by immune cells, the most abundant of which in thyroid cancer are tumour-associated macrophages (TAMs). High TAM density has been indeed associated with larger tumour size and lymph node metastases in PTC [5,6] as well as with high histologic grade and invasiveness in poorly differentiated thyroid cancer (PDTC) [7,8]. Moreover, in both PTC and PDTC, TAM density is correlated with reduced survival [5,7]. The correlation between increasing presence of TAMs and cancer progression and lethality has also been confirmed in ATC, where TAMs could represent more than 50% of the tumour mass [9], although the molecular bases of this correlation are still unclear. It has been shown that advanced thyroid cancer presents massive TAMs infiltration [8,9] and that defined macrophage subpopulations play different roles in determining the aggressive cancer course [10,11,12]. Infiltrating TAMs could, indeed, belong to the tumour-suppressive (M1) or the pro-tumourigenic (M2) subtype [13]. It has been demonstrated that the increase in M2 macrophages during thyroid cancer progression contributes to cancer aggressiveness by promoting angiogenesis and immune escape, particularly in BRAF-mutated cancer [8,10].

Recently, several polymorphisms in regulatory and/or coding regions of the transcription factor *FOXE1* have been associated with increased susceptibility to PTC [14,15] and aggressive forms of thyroid cancer [16,17]. In particular, polymorphisms that increase *FOXE1* expression levels have been associated with several bad clinical parameters such as severe histopathological characteristics, reduced degree of differentiation, invasion of lymph nodes and metastases [16,18]. By using a conditional mouse model of BRAF^V600E^-inducible thyroid cancer [19] crossed with *FOXE1* heterozygous knockout mice, we recently demonstrated that *FOXE1* gene dosage affects cancer histology, proliferation and differentiation [20], thus showing for the first time a cause–effect relationship between *FOXE1* and the thyroid cancer phenotype. To investigate the gene expression programme induced by *FOXE1*, here we ectopically express it in FRT rat thyroid cells lacking endogenous *FOXE1* expression, showing that the differentially expressed genes are particularly enriched in categories related to immune cells functions. Moreover, *FOXE1*-expressing cells increase their migratory ability, being also able to induce monocytes chemotaxis. We then checked if *FOXE1* was also involved in a similar phenotype in vivo in a thyroid cancer model with heterozygous *FOXE1* knockout. Consistent with in vitro data, reduction of FOXE1 gene dosage downregulates the expression of immune-related genes by reducing Tam’s recruitment and impairing their polarisation towards anti-inflammatory and pro-tumourigenic M2 status.

Our results highlight a *FOXE1* function never described before, which goes beyond the regulation of thyroid-specific molecules and reveals a novel pathway of modulation of macrophage infiltration in the thyroid cancer microenvironment.

## 2. Results

### 2.1. FOXE1 Expression Induces a Migratory Phenotype in FRT Cells

Cells in the Fischer rat thyroid epithelial cell line (FRT) are poorly differentiated follicular cells derived from Fischer rat thyroid glands that do not express FOXE1 or other thyroid differentiation markers, except for PAX8 [21]. FOXE1-expressing FRT cells were generated by transfection with a FOXE1-expressing vector. FOXE1 RNA and protein expression were measured in three independent FOXE1 clones (FC) by RT-PCR and Western blot, respectively, observing that FC26, FC19, and FC55 expressed different FOXE1 protein levels. FRT parental cells (from here on reported as ctr) are negative for FOXE1 expression, as expected (Figure 1A). FOXE1 nuclear localisation was confirmed by immunofluorescence (Figure 1B), revealing that the FC55 clone shows the most homogeneous staining. As FOXE1 induces thyroid cell motility both in development and in cancer [22,23], we analysed the migratory behaviour of FC. Cell motility was then measured by transwell assay: FOXE1-expressing clones and control cells were seeded in the upper transwell chamber in serum-free medium and allowed to migrate. Representative images of the transwell before (total cells) and after (migrating cells) membrane cleaning is reported in Figure 1C, and the quantification of migrated cells per analysed area is reported in Figure 1D, showing a significant increase of cell migration only for the clones FC19 and FC55. These results suggest that FOXE1, at high expression levels, stimulates FRT cell motility.

### 2.2. Transcriptomic Landscape of FOXE1-Expressing FRT Cells

To investigate the transcriptional programme induced by FOXE1 in thyroid epithelial cells, we selected the FC55 displaying the highest and most homogeneous expression of FOXE1. RNAseq analysis was performed on total RNA from FC55 and control cells. Differentially expressed genes (DEG) were considered with a ±2-fold change and *p* ≤ 0.05 cut-off and analysed by Gene Ontology (GO). A total of 750 DEG resulted as being regulated ± 2-folds, of which 469 (62.5%) were upregulated and 281 (37.5%) were downregulated (Figure 2A). GO terms enriched in upregulated genes were clustered in functional categories mainly related to signal transduction, macrophage recruitment and differentiation, metal ion homeostasis and transport and extracellular matrix components (Figure 2B). GO terms significantly enriched in downregulated genes were instead mainly related to cell cycle, epigenetic regulation of chromatin, immune cell migration and cholesterol metabolism (Figure 2C). A panel of DEG, grouped for functional categories, was then validated performing quantitative RT-PCR on the three clones FC26, FC19 and FC55 (Appendix A).

### 2.3. FOXE1 Enhances Chemoattractant Ability of FRT Cells

The observed enrichment of multiple functional categories related to immune cells recruitment, migration and differentiation in *FOXE1*-induced DEG was largely unexpected. Particularly, several genes involved in macrophages chemotaxis and differentiation were clustered in the upregulated genes list. Among these, the most strongly upregulated was *CCL2*, a well-established macrophage chemoattractant. In addition, several other genes encoding secreted proteins involved in macrophage recruitment and activation, such as *CCL7, CSF1, LGALS3BP* and *INPP5D* [24,25,26], resulted as being upregulated in FC55 by RNA-seq (Figure 3A). Upregulation of these macrophage ligands was validated by quantitative RT-PCR in the FC, revealing a correlation between *FOXE1* dosage and the extent of these genes’ upregulation (Figure 3B). To investigate if the induction of these genes could change macrophage recruitment by chemoattractants release in the extracellular medium, we tested the ability of FC to induce monocyte chemotaxis by co-culture assay. FRT and the three *FOXE1* clones were cultured in the transwell lower chamber in order to obtain secreted proteins accumulation in the medium. U937 human monocytes were then added in the upper chamber in serum-free medium, and their migration through the transwell porous membrane was measured (Figure 3C). A statistically significant increase of U937 cells migration was observed only when co-cultured with the high *FOXE1* expressing clones FC19 and FC55 (Figure 3D), demonstrating a *FOXE1* dose-dependent gain in chemoattractant activity.

### 2.4. Chemokine Expression and Immune Cell Recruitment Are FOXE1-Dependent in an In Vivo Thyroid Cancer Model

It has been demonstrated that conditional activation of oncogenic BRAF in mouse results in the induction of thyroid cancers with increased CCL2 and CSF1 expression, associated with the recruitment of tumour-associated macrophages (TAMs) [12]. Given that FOXE1 expression induces both CCL2 and CSF1, together with other chemoattractants, in thyroid cells in vitro, we asked if FOXE1 could modulate the oncogenic induction of chemokines and hence the infiltration of thyroid cancer by TAMs in vivo. To answer this question, we used a FOXE1^+/+^ and FOXE1^+/-^ BRAF^V600E^-dependent thyroid cancer mouse model (BRAF) to induce thyroid cancer in the presence of reduced FOXE1 expression [20]. Mice were treated with doxycycline for one week to develop thyroid cancer [19], and the expression levels of CCL2, CSF-1, LGALS3BP, INPP5D and CCL7, together with that of the chemokine receptors CCR2 and CSF1R, were measured in cancer specimens by quantitative RT-PCR. Tumours developed in BRAF FOXE1^+/-^ background showed significantly reduced expression of all the genes investigated compared with those developed in BRAF FOXE1^+/+^ mice (Figure 4A,B). In this cancer model, it is known that BRAF^V600E^ activation induces a robust recruitment of M2-polarised TAMs [12]. Tumour infiltration by TAMs was then evaluated on thyroid cancer sections by immunohistochemistry (IHC) for the macrophage markers IBA1 and for GALECTIN-3, this latter being described to regulate alternative macrophage activation towards an M2 pro-tumourigenic state [27,28,29]. As shown in Figure 4C,D, both markers show reduced staining in the BRAF FOXE1^+/-^ cancer samples with respect to the BRAF FOXE1^+/+^ ones. The expression of two additional M2-related markers, ARGINASE1 (ARG) and INTERLEUKINE10 (IL10) [30,31], was measured to further evaluate if the reduction of FOXE1 levels could affect the abundance of this macrophage subpopulation within the tumour. BRAF FOXE1^+/-^ cancers showed a weaker expression of both M2 markers with respect to the BRAF FOXE1^+/+^ ones (Figure 4E). Since cancer-associated fibroblasts (CAFs), together with TAMs, are the main components of the tumour stroma, we looked at α-SMA expression as a CAFs-specific marker. IHC for α-SMA reveals that reducing FOXE1 levels in BRAF^V600E^-induced thyroid cancer decreases the CAFs stromal expansion observed in FOXE1 wild type background (Figure 4F). As control, we analysed WT and FOXE1^+/-^ mice, that do not develop cancer because they are not transgenic for BRAF^V600E^ [20], observing no statistically significant differences in the expression of chemokines and their receptors (Appendix A) or in IHC staining for IBA1 (Appendix A).

## 3. Discussion

The transcription factor FOXE1, essential for thyroid organogenesis and function, has been identified as susceptibility gene for differentiated thyroid cancer. Most of cancer-associated *FOXE1* polymorphisms are likely to regulate the gene’s expression, as the polymorphisms fall in locus regulatory regions. Despite the identification of numerous *FOXE1* variants associated with PTC risk, it is poorly understood how they modulate thyroid cancer susceptibility at the molecular level. Recently, we demonstrated that *FOXE1* dosage alteration affects thyroid cancer histology, proliferation and differentiation in vivo [20], thus showing for the first time a cause–effect relationship between *FOXE1* and thyroid cancer phenotype. To further investigate the phenotypic effects of *FOXE1* level variation on thyroid cancer, here we combined two complementary approaches: gain-of-function in vitro and loss-of-function in vivo. The transcriptional landscape induced by *FOXE1* in FRT thyroid cells in vitro is enriched in differentially expressed genes involved in immune response and macrophage recruitment/activation, such as the chemokines *CCL2, CSF1, LGALS3BP, INPP5D* and *CCL7*. *FOXE1*-expressing cells increase their ability to induce monocytes chemotaxis in vitro, thus indicating that the upregulated chemokines are secreted and functional. Interestingly, these genes result frequently in being upregulated during the progression of several cancers [32,33,34]. The *BRAF*^V600E^-dependent mouse model of thyroid cancer with a single functional *FOXE1* allele (*FOXE1*^+/-^) was then used to test if the identified genes were under *FOXE1* control, also in vivo. When cancer was induced, reduced expression of the identified chemokines was observed in *FOXE1*^+/-^, thus showing that these genes are under FOXE1 control, being both upregulated in vitro by *FOXE1* ectopic expression and downregulated in vivo by *FOXE1* hemizygosity. It is worth noting that such correlation does not occur in a normal thyroid, being evident only in poorly differentiated and/or neoplastic contexts, such as FRT cells and BRAFV600E-induced cancer, respectively, similar to what has already been demonstrated for *FOXE1* control of thyroid differentiation [20]. Because *FOXE1*-regulated chemokines CCL2 and CSF1, through the binding of the myeloid cells receptors CCR2 and CSF1R, respectively, are responsible for TAMs recruitment in several solid cancers [12,35,36], we evaluated TAMs infiltration in the experimental thyroid cancers by measuring the expression of the two receptors and the TAM markers IBA1 and GALECTIN-3, involved in macrophages motility/migration [28,29]. Our results show that reduction in vivo of FOXE1 impairs the upregulation of both IBA1 and GALECTIN-3, induced by BRAFV600E activation, indicating a FOXE1-dependent reduction of TAMs recruitment. In particular, the decrease of GALECTIN-3 staining, ARGINASE and IL10 levels indicates that this reduction mainly concerns the M2 population. It has already been shown, in the same cancer model, that targeting *CLC2* and *CSF1* impairs TAMs recruitment and M2 polarisation [12], as we here obtained by targeting *FOXE1* as well. This suggests that in thyroid, the processes driven by these chemokines could be downstream of *FOXE1*, indicating that acting on *FOXE1* could affect at the same time *CCL2/CCR2* or *CSF1/CSF1R* signalling that are currently faced individually [37,38,39].

TAMs are important players in cancer progression because of their specific function and their contribution in modelling and determining cancer stroma properties [40] together with CAFs. Consistently, we also observed a reduction of CAFs in *BRAF FOXE1*^+/-^ tumour stroma. CAFs can promote tumourigenesis and metastasis dissemination by sustaining inflammation, angiogenesis, immunosuppression, epithelial–mesenchymal transition and extracellular matrix remodelling [41,42]. Thus, the crosstalk between CAFs and M2 macrophages and their synergistic effect on cancer progression [43] could be impaired in *FOXE1*^+/-^ cancers. The demonstration that variations in *FOXE1* dosage change the tumour microenvironment suggests a previously unsuspected function for this transcription factor in addition to the well-established regulation of thyrocyte differentiation, indicating that FOXE1 could play a role more in thyroid cancer progression than in its onset. In conclusion, this study highlights a novel function of *FOXE1* in modulating TAMs infiltration and stromal organisation in differentiated thyroid cancer. This is the first time, to the best of our knowledge, that *FOXE1* has been connected to the expression of immune-related genes, thus paving the way to novel perspectives to understand the contribution of such a transcription factor to thyroid cancer susceptibility.

## 4. Materials and Methods

### 4.1. Plasmid Preparation

Rat *FOXE1* CDS sequence (NM_138909.1) was amplified by PCR from the pCMV *FOXE1* plasmid [44] and subcloned in the NheI site of CET 1019 AS-puro-SceI plasmid (Merck Millipore Corporation, Burlington, MA, USA).

### 4.2. Cell Culture and Transfection

FRT (Fisher rat thyroid cells) cells were cultured in Coon’s Modified Ham’s F12 Medium (Euroclone, Milan, Italy) supplemented with 5% foetal bovine serum (Euroclone), 1% penicillin (Euroclone) and 1% streptomycin (Euroclone). U937 cells were cultured in RPMI supplemented with 10% foetal bovine serum. FRT cells were stably transfected with CET 1019 AS-Puro-SceI- *FOXE1* plasmid using FuGene 6 Transfection Reagent (Roche, Basel, Switzerland) according to manufacturer’s instruction. Briefly, 3 μg of CET 1019 AS-Puro-SceI-*FOXE1* plasmid or 3 μg of empty vector was transfected in FRT cells seeded in 100 mm culture dishes (Corning, NY, USA) at 20% confluence. Forty-eight hours after transfection, cells were selected in complete medium supplemented with 1 μg/mL of puromycin. After two weeks of continuous selection, single colonies were picked from plates transfected with either CET 1019 AS-Puro-SceI-*FOXE1* vector or empty vector. The clones overexpressing the greatest amount of *FOXE1* (*FOXE1*-FRT) were used for the showed experiments.

### 4.3. RNA Sequencing and Bioinformatic Analysis

Total RNA from FRT and *FOXE1*-FRT was assessed with Agilent 2100 Bioanalyzer (Agilent technologies, Santa Clara, CA, USA). Indexed libraries were obtained from 500 ng of each sample using TrueSeq polyA+ mRNA Sample Prep Kit (Illumina, Sam Diego, CA, USA) according to manufacturer’s instructions. Three replicates for each sample were analysed. HiSeq1500 platform (Illumina) was used to perform libraries sequencing at 8 pmol/ul per lane (paired-end, 2 × 100 cycles) with coverage of about 20 × 106 reads per sample. FastQC (http://www.bioinformatics.babraham.ac.uk/projects/fastqc/ (accessed on 10 June 2021)) was used to control sequencing raw data (.fastq files), and the selected reads were aligned to rat genome (rn4/Baylor 3.4 assembly) using Tophat 2.0.10 [45] with standard parameters. Gene annotation was obtained for all known genes in the rat genome as provided by UCSC (rn4) (https://support.illumina.com/sequencing/sequencing_software/igenome.ilmn (accessed on 10 June 2021)). To identify differentially expressed genes between FRT and *FOXE1*-FRT, the software HTSeq-count [46] was used to find known genes with more than 10 reads that served as input of DESeq software 1.14.0 [47] to normalise each transcript expression value. Transcripts with a fold change <−3 or >3 and a *p*-value < 0.05 were selected. Differentially expressed genes functional annotation analysis was performed by using DAVID. The web-based clustering tool was used for clustering different functional annotation categories: gene ontology, pathways, protein interaction, protein domains, tissue expression, sequences features and keywords. Only clusters containing more than one term with *p* < 0.05 were considered. The raw data are available as supplementary files.

### 4.4. Immunoblotting

Whole cell lysates of FRT cells or stable *FOXE1* clones were prepared using 50 mM Tris HCl pH 8, 5 mM MgCl_2_, 150 mM NaCl, 0.5% Deoxycholic Acid, 0.1% SDS, 1% Triton, 1× protease inhibitor cocktail, 0.5 mM PMSF, 5 mM sodium orthovanadate (Na_3_VO_4_), 10 mM sodium fluoride (NaF), 0.5 mM sodium pyrophosphate (Na_4_P_2_O_7_) and 1 mM dithiothreitol (DTT) (all provided by Sigma-Aldrich, St. Luis, MO, USA). Cell lysates were incubated 15 min on ice and centrifuged at full speed at 4 °C for 25 min. Protein concentration was estimated using the BCA Protein Assay Kit (Thermo Fisher Scientific Inc., Rockford, IL, USA).

Protein lysates were loaded on precasted NuPAGE 4–12% Bis-Tris gels (Life Technologies Ltd., Carlsbad, CA, USA). Gels were electroblotted on Immobilion-P PVDF membranes (Millipore, Billerica, MA, USA) and screened for FOXE1 (De Vita et al. 2005) and TUBULIN (Sigma-Aldrich).

Secondary antibodies mouse IgG horseradish peroxidase-linked whole antibody (GE Healthcare, Little Chalfont, Buckinghamshire, UK) and rabbit IgG horseradish peroxidase-linked whole antibody (GE Healthcare) were used as indicated by manufacturers. Chemiluminescence was detected using Pierce ECL Western blotting substrate (Thermo Fisher Scientific Inc., Rockford, IL, USA).

### 4.5. Immunofluorescence Assay

Cells were cultured on 12 mm diameter glass coverslips. At 24 h after seeding, cells were fixed in 4% PFA for 15 min, permeabilised for 3 min in 0.2% Triton X-100 and blocked for 30 min in 5% BSA. Cells were incubated for 1 h at RT with primary antibody anti-FOXE1, washed thrice with PBS 1X and incubated after with secondary FITC-tagged goat anti-rabbit antibody (Jackson Immunoresearch Laboratories Inc, West Grove, PA, USA). Immunofluorescence was captured by confocal laser scanner microscope Zeiss LSM 510 (Carl Zeiss AG, Oberkochen, Germany).

### 4.6. Transwell Assay

Transwell Permeable Support with 8.0 μm (for FRT) or 5.0 μm (for U937) polycarbonate filter membrane 6.5 mm insert provided by Corning, were used for motility assay. To evaluate *FOXE1* clones’ migration, cells were then fixed for 15 min in PFA 4% permeabilised with 0.2% Triton X-100 for 5 min, blocked with 5% BSA for 30 min and stained with Dapi for 5 min. The results of two independent experiments performed in duplicate are reported; for each replicate, a minimum of 4 areas were analysed. To evaluate U937 migration, FRT and *FOXE1*-FRT were seeded in transwell lower chamber to allow cell growth and accumulation of secreted proteins in culture medium for 48 h, then U937 were added in the upper chamber, and their migration was evaluated 4 h after. Representative areas of the transwell were acquired with a 10× objective. Cells migrated in the lower chamber were recovered from the suspension and counted; the results of two independent experiments performed in triplicate are reported. For each replicate, migrating cells were counted twice using a Neubauer chamber. Representative areas of the transwell were acquired with a 10× objective.

### 4.7. Quantitative Real-Time PCR

Total RNA isolation and retro-transcription were performed as described in Credendino et al. 2020 [20]. Quantitative Real-Time PCR on total cDNA was performed with iTaq Universal SYBR Green Supermix (Bio-Rad. Hercules, California, USA) using gene-specific oligonucleotides. Mouse *CSF1, CSF1R, CCL2, CCL2R, ARGINASE, IL10* and *ß-ACTIN* oligonucleotides were from Ryder et al. [12]. RAT *CCL2, RAT CSF1, CCL7. LGALS3B, INPP5D* and *TUBULIN* oligonucleotides sequences are:

RAT *CCL2* FW: TGTAGCATCCACGTGCTGTC

RAT *CCL2* REV: CCGACTCATTGGGATCATCT

RAT *CSF1* FW: CTGCCCTTCTTCGACATGGAT

RAT *CSF1* REV: CTACAGTGCTCCGACACCT

*CCL7* FW: TGAAGCCAGCTCTCTCTCTC

*CCL7* REV: TGGATGAATTGGTCCCATCT

*LGALS3B* FW: CTCTGTGTTCTTGCTGGTTCC

*LGALS3B* REV: CTGAGGCCCCATTAACCA

*INPP5D* FW: ACCTGCAGTTCCCCGTGC

*INPP5D* REV: GGTGGGCATGACACTTTCTG

*TUBULIN* FW CAACACCTTCTTCAGTGAGACAGG

*TUBULIN* REV: TCAATGATCTCCTTGCCAATGGT

### 4.8. Mice Treatment and Genotyping

Animals were kept, fed and genotyped as already described [20].

### 4.9. Immunohistochemistry

Thyroid sections were obtained and stained as described in Credendino et al. 2020 using the following primary antibodies: GALECTIN-3 (#CL8942B, Cedarlane, Burlington, Canada), alpha smooth muscle actin (α-SMA) (#M0851, Agilent-Dako, Santa Clara, California, USA) and IBA-1 (#019-19741, Wako, Neuss, Deutschland). Images were obtained as described in Credendino et al. 2020 [20].

## Figures and Tables

**Figure 1 ijms-22-07666-f001:**
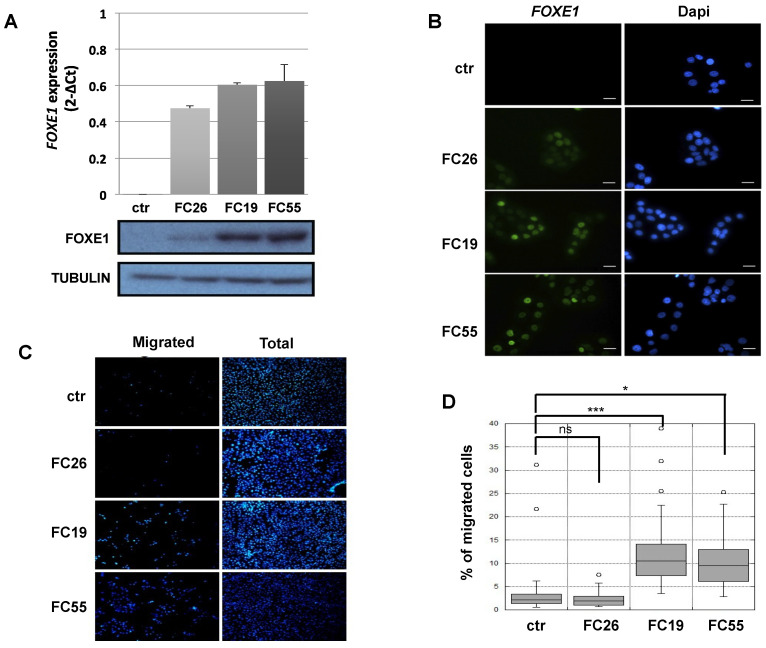
FOXE1 elicits thyroid cell migration of FOXE1-overexpressing clones. (**A**) FOXE1 expression in ctr cells and different FOXE1 clones (FC) analysed by qRT PCR (upper panel) and Western blot (bottom panel). Tubulin was used to normalise. (**B**) Immunofluorescence images of the same clones as in A, probed with anti-FOXE1 antibody, showing nuclear FOXE1 localisation. Nuclei are stained with Dapi. 40× magnifications are shown. The scale bar represents 20 um. (**C**) Transwell assay performed on control cells (ctr) and FOXE1-expressing clones FC26, FC19, FC55. Representative images of the transwell used before (total) and after (migrated) swab cleaning of the inner part of the upper chamber. Cells were stained with Dapi in order to visualise the nuclei. (**D**) The percentage of migrated on total cells was calculated. One-way ANOVA test was performed on data, and *p*-values are reported for each clone relative to ctr. * *p* < 0.05, *** *p* < 0.001, ns: not significant.

**Figure 2 ijms-22-07666-f002:**
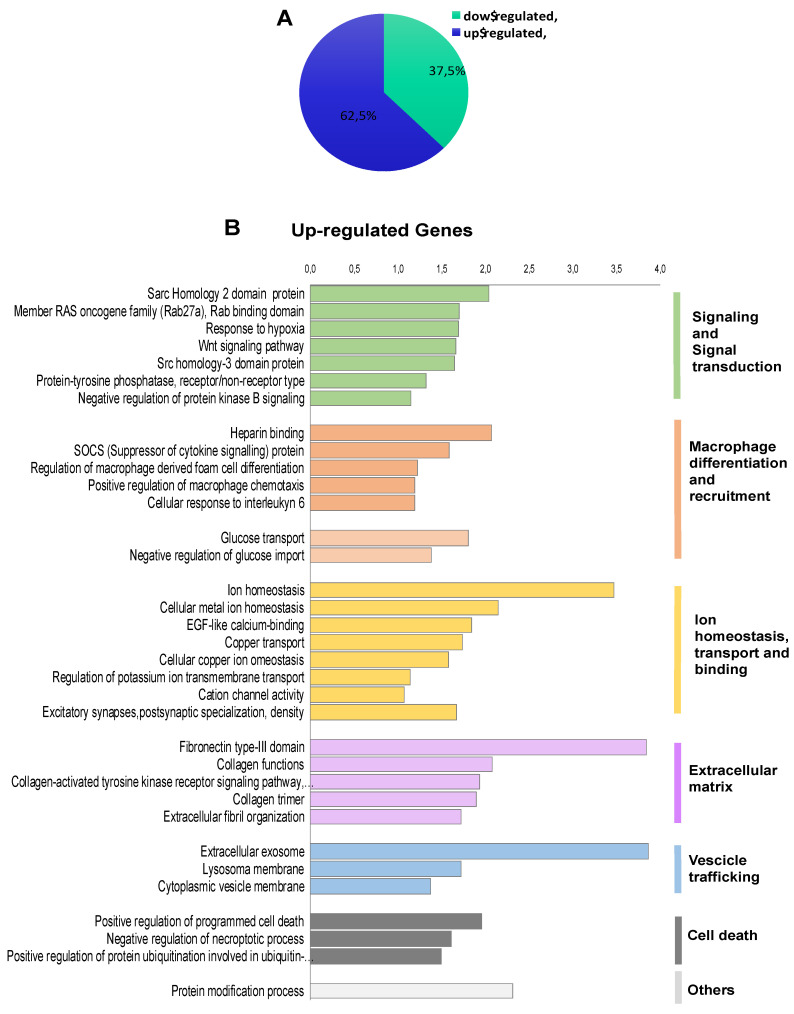
Functional annotation of differentially expressed genes in FOXE1-overxpressing cells. RNAseq analysis was performed on total RNA from FC55 and control cells. (**A**) Pie chart of differentially expressed genes (DEG), considered with a ±2-fold change (FC) and *p* ≤ 0.05 cut-off and analysed by Gene Ontology (GO). A total of 750 DEG resulted as being regulated, of which 469 (62.5%) upregulated and 281 (37.5%) downregulated. Functional categories were significantly enriched among genes that were either (**B**) upregulated or (**C**) downregulated. Functional categories (highlighted on the right in vertical groups) were obtained by clustering the functional annotation contained in the web-based DAVID database. To perform clustering, we used multiple annotations—namely, gene ontology for biological processes, molecular function and cellular components (GO_DIRECT), pathways, protein interactions, protein domain and tissue expressions. Enrichment scores are plotted only for clusters containing multiple terms with *p* < 0.05.

**Figure 3 ijms-22-07666-f003:**
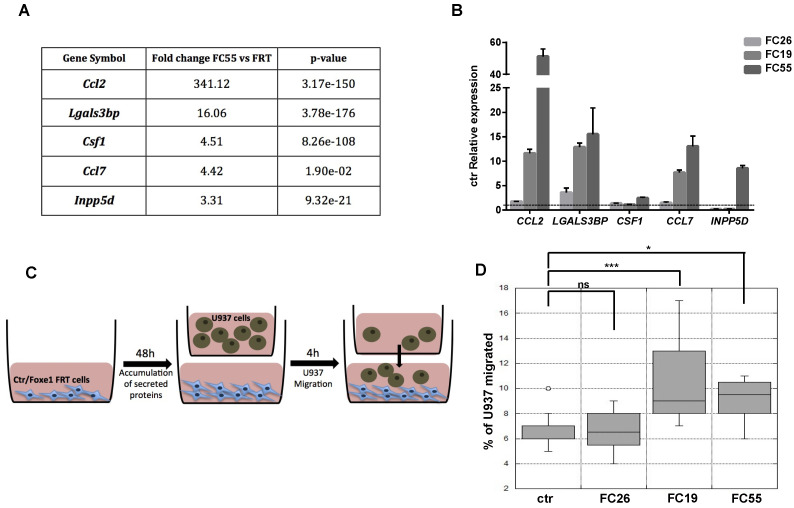
FOXE1 modulates macrophage-related genes and their migration in vitro. (**A**) Macrophage-related genes from RNA-seq analysis of FOXE1-expressing FRT. (**B**) Quantitative RT-PCR analysis of genes in A in FOXE1 clones with low (FC26), moderate (FC19) and high (FC55) FOXE1 expression. The data are shown as ctr-relative expression (2^-∆∆Ct^). Actin was used to normalise. The starting value of the control, set to 1, is indicated by the dotted line. (**C**) Ctr cells and FOXE1 clones were seeded in transwell low chamber and U937 cells were added in the upper chamber after 48 h. (**D**) The percentage of migrating U937 cells was evaluated after 4 h. One-way ANOVA test was performed on data, and *p*-values are reported for each clone relative to ctr. * *p* < 0.05, *** *p* < 0.001, ns: not significant.

**Figure 4 ijms-22-07666-f004:**
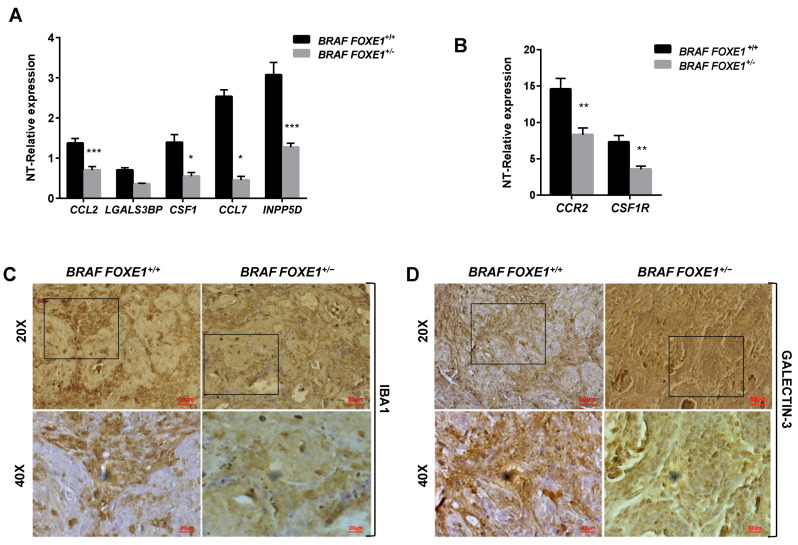
FOXE1 modulates macrophage recruitment in vivo. (**A**) *BRAF FOXE1*^+/+^ and *BRAF FOXE1*^+/-^ mice were treated with doxycycline for one week to induce *BRAF*^V600E^-dependent thyroid cancer. RNA was isolated from pools of thyroids with the same genotype and treatment. Macrophage-related genes and (**B**) *CCR2* and *CSF1R* expression was analysed by quantitative RT-PCR, and the results are reported as the fold change of *BRAF FOXE1*^+/+^- and *BRAF FOXE1*^+/-^-treated groups on the respective untreated group. * *p* < 0.05, ** *p* < 0.01; *** *p* < 0.001. (**C**) Thyroid sections were analysed by immunohistochemical staining for macrophage markers IBA1 and (**D**) GALECTIN3. Nuclei are stained with haematoxylin. 20× (upper panels) and 40× (lower panels) of the 20× boxed area, magnifications are shown. (**E**) Expression analysis of M2 macrophage subpopulation markers *ARGINASE1* (ARG) and *INTERLEUKINE10* (IL10) on the same samples as in A. * *p* < 0.05. (**F**) Immunohistochemical analysis of α-SMA in thyroid sections from *BRAF FOXE1*^+/+^ and *BRAF FOXE1*^+/-^ doxycycline-treated mice. Nuclei are stained with haematoxylin. 20X (upper panel) and 40X (lower panels) of the area boxed in 20× magnifications are shown. The images are representative of six different images.

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
