# Peer review of "FOXE1-Dependent Regulation of Macrophage Chemotaxis by Thyroid Cells In Vitro and In Vivo"

_ijms, 2021, doi:10.3390/ijms22147666_

Round 1

Reviewer 1 Report

  1. It is not clear how many RNA-seq samples(control and FC55) the researcher used to do the DEG analysis.
  2. Since FOXE1 was also highly expressed in clone FC19, are you able to identify the same set of DEG genes and enriched terms by comparing the control and FC19 clone? Although a panel of DEG has been validated by RT-PCR, the audience could be interested in the replication of the work in terms of gene expression systematically.
  3. In both DEG analysis and GO analysis, p-values were used which could raise the concern of the multiple-testing problem.
  4. Ignore all grammar suggestion in Figure 3.A
  5. Another kind reminder for future analysis is that Tophat2 has been gradually deprecated and can be replaced by HISAT2. 

Overall, this article reveals a novel path through FOXE1 and would benefit the current research of thyroid cancer.

Author Response

  1. It is not clear how many RNA-seq samples (control and FC55) the researcher used to do the DEG analysis.

We performed RNA-seq analysis on three replicates for each sample (control and FC55). We added this information in Materials and Methods section.

  1. Since FOXE1 was also highly expressed in clone FC19, are you able to identify the same set of DEG genes and enriched terms by comparing the control and FC19 clone? Although a panel of DEG has been validated by RT-PCR, the audience could be interested in the replication of the work in terms of gene expression systematically.

We agree with the reviewer that the replication of the systematic gene expression analysis on another clone would be an interesting point, although in this paper we did not focused on this aspect because we used the gene expression profile as the starting point for the identification of FOXE1-dependent cancer-related cellular phenotypes.

  1. In both DEG analysis and GO analysis, p-values were used which could raise the concern of the multiple-testing problem.

GO terms that were used for the clustering analysis had all FDR values ≤0.1. Multiple-testing correction was not taken into account after clustering, but few clusters were validated experimentally (as shown in fig S1).

  1. Ignore all grammar suggestion in Figure 3.A.

Done.

  1. Another kind reminder for future analysis is that Tophat2 has been gradually deprecated and can be replaced by HISAT2.

We thank the reviewer for the useful reminder: for future analysis we will consider the use of HISAT2.

Reviewer 2 Report

The Authors present an interesting paper connecting FOXE1 expression and the development of thyroid cancer. However some issues should be addressed: it is not clear to me the role of FOXE1 in TAM regulation and tumor promotion,since in vitro and in vivo data suggest that at least in vivo FOXE1 loss induces a reduction in macrofage activation, but still the cancer is activated, whereas in vitro overexpression of FOXE1 activates the transcription of these genes..so how do they connect these data? It would be interesting to investigate in vivo the overexpression of FOXE1.

Other points are the following: 

1) in figure 1D an ANOVA test would be more suitable than a t-test for significance. 

2) for the RNAseq analysis, are the p-value reported  after FDR correction? How many replicas for each cell line were performed?

3) in figure 3D which statistical analysis has been carried out?

I think these issues should be addressed before publication of this interesting paper

Author Response

The Authors present an interesting paper connecting FOXE1 expression and the development of thyroid cancer. However some issues should be addressed: it is not clear to me the role of FOXE1 in TAM regulation and tumor promotion, since in vitro and in vivo data suggest that at least in vivo FOXE1 loss induces a reduction in macrofage activation, but still the cancer is activated, whereas in vitro overexpression of FOXE1 activates the transcription of these genes.so how do they connect these data? It would be interesting to investigate in vivo the overexpression of FOXE1.

Thanks to the reviewer for this observation. Our data show that FOXE1 levels positively correlate with the expression of the same set of chemokines. These are indeed induced in vitro by its overexpression and downregulated in vivo by its reduction. Consequently, FOXE1 overexpression increases the ability of thyroid cells to attract macrophages in vitro, while its reduction decreases thyroid cancer infiltration by macrophages in vivo. These data obtained in two different models reinforce and complement very well each other. The resulting information is that FOXE1 is positively involved in the production of chemoattractants by thyroid cells. As macrophage infiltration is higher in more aggressive cancers, our data suggest that FOXE1 is likely to play a role in the progression of thyroid cancer rather than in its onset.  Indeed, we recently published that heterozygous FOXE1 knockout does not interfere with cancer induction by Braf, but instead with its differentiation and proliferation (Credendino et al, 2020, ref n. 20).  To be more clear on these issues, we modified several sentences in the discussion, starting from lane 216.

Other points:

  1. in figure 1D an ANOVA test would be more suitable than a t-test for significance.

We thank the reviewer for the suggestion. In the revised version, we performed One Way ANOVA test on data and the resulting significance is now reported in figure 1D and its legend.

  1. for the RNAseq analysis, are the p-value reported after FDR correction? How many replicas for each cell line were performed?

The p-value for the RNA-seq analysis are reported after FDR correction using an FDR value ≤0.1. Three replicates for each sample (control and FC55) were analyzed. We added the missing information in Materials and Methods section.

  1. in figure 3D which statistical analysis has been carried out?

In the original figure 3D, t-test results are reported. As the Reviewer suggested for figure 1D, we performed One Way ANOVA test on these data and the resulting significance has been substituted in figure 3D and its legend.

Round 2

Reviewer 2 Report

I have no more comments. I think the manuscript deserves publication. One minor poit is that at line 227 maybe the Authors need to insert a space between the two words "severalcancers".